# SELECT2REASON: EFFICIENT INSTRUCTION-TUNING DATA SELECTION FOR LONG-COT REASONING

## ABSTRACT

A practical approach to activate long chain-of-thoughts reasoning ability in pre-trained large language models is to perform supervised fine-tuning on instruction datasets synthesized by strong Large Reasoning Models such as DeepSeek-R1, offering a cost-effective alternative to reinforcement learning. However, large-scale instruction sets with more than 100k samples incur significant training overhead, while effective strategies for automatic long-CoT instruction selection still remain unexplored. In this work, we propose SELECT2REASON, a novel and efficient instruction-tuning data selection framework for long-CoT reasoning. From the perspective of emergence of rethinking behaviors like self-correction and backtracking, we investigate common metrics that may determine the quality of long-CoT reasoning instructions. SELECT2REASON leverages a quantifier to estimate difficulty of question and jointly incorporates a reasoning trace length-based heuristic through a weighted scheme for ranking to prioritize high-utility examples. Empirical results on OpenR1-Math-220k demonstrate that fine-tuning LLM on only 10% of the data selected by SELECT2REASON achieves performance competitive with or superior to full-data tuning and open-source baseline OpenR1-Qwen-7B across three competition-level and six comprehensive mathematical benchmarks. Further experiments highlight the scalability in varying data size, efficiency during inference, and its adaptability to other instruction pools with minimal cost.

## 1 INTRODUCTION

Large reasoning models (LRMs) (OpenAI, 2024; Guo et al., 2025; DeepMind, 2025), mark a significant leap in the complex reasoning abilities of large language models (LLMs). With the emergence of the long chain-of-thoughts (long-CoT) reasoning ability (Chen et al., 2025a), these models exhibit human-like behaviors such as exploration, verification, reflection, and correction, allowing them to autonomously derive multi-branch and multi-step solutions via deliberate planning and backtracking (Huang & Chang, 2022; Li et al., 2025c).

A practical approach to activate long-CoT reasoning ability in pre-trained LLMs is to perform supervised fine-tuning (SFT) on instructions synthesized by strong LRMs. Open-source projects (Face, 2025; Team, 2025; Liu et al., 2025) release over 100K such instructions respectively, yet large-scale SFT still entails significant costs. Recent work argues that the *quality* of long-CoT data, rather than *quantity* is more critical. For example, LIMO (Ye et al., 2025) applies multiple rounds of sampling and filtering over tens of millions of problems and employs expert-designed solutions to curate a compact yet high-quality dataset of 817 samples. Similarly, s1 (Muennighoff et al., 2025) depends heavily on API models and intricate data engineering pipelines tailored to optimize for quality, difficulty, and diversity, yielding 1k examples. Unfortunately, their metrics are based on qualitative heuristics without rigorous quantitative validation, and these carefully-curated pipelines are often not publicly available which impedes reproducibility and generalization.

Recently, research on instruction selection (Chen et al., 2023; Liu et al., 2023b; Lu et al., 2023; Zhang et al., 2024c; Yang et al., 2024c; Li et al., 2023a; Liu et al., 2024) has explored various aspects of data quality to automatically extract high-utility subsets from large instruction pools. However, the specific challenge of **instruction selection for long-CoT reasoning** remains largely unaddressed. We investigate the features that may determine the quality of long-CoT instructions. The emergence of rethinking behaviors in long-CoT traces is regarded as an **aha moment** for

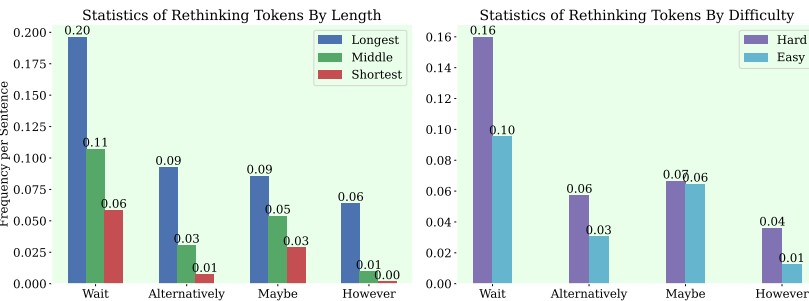

Figure 1: Statistics of rethinking tokens in reasoning trace. Longer reasoning traces exhibit a higher frequency of rethinking tokens in each step such as *Wait*, *Alternatively*, *Maybe*, *However*, which also occurs often in instruction with questions that are hard to solve.

LRMs (Guo et al., 2025), indicating that the model learns to allocate more inference time toward self-correction and backtracking. Previous studies (Xie et al., 2025; Yeo et al., 2025) use the frequency of rethinking-related keywords as a proxy for reasoning quality, serving as a signal of the model's internal thinking patterns. Similarly, s1 (Muennighoff et al., 2025) implements budget forcing by appending extrapolation strings like *Wait* to extend thinking process. We hypothesize that reasoning traces exhibiting more rethinking behaviors may serve as higher-quality instructions and offer greater training value. However, only qualitative keyword-based metrics cannot fully capture the complexity of reasoning patterns (Zeng et al., 2025), highlighting the need for quantitative evaluation metrics.

We investigate common metrics that may lead to higher frequency of rethinking tokens in long-CoT reasoning trace, and statistical analysis presented in Figure 1 reveals that **longer** reasoning traces exhibit more rethinking tokens in each step such as *Wait*, *Alternatively*, *Maybe* and *However*, which also occurs often in instruction with questions those are **hard** to solve. According to results in Figure 3, models fine-tuned on subsets prioritized by the longest reasoning traces consistently outperform those trained on the middle or shortest traces across various data scales. It can be concluded that the **length of the reasoning trace** in the response is a simple but tough-to-beat heuristic for selection. Furthermore, models trained on instruction subsets which are hard to solve by base model significantly outperform those trained on subsets with easy questions, aligning with the intuition in (Ye et al., 2025; Muennighoff et al., 2025) that more challenging instructions provide greater learning value. However, the challenge of **automated, difficulty-aware** instruction selection remains largely unaddressed.

To this end, we propose **SELECT2REASON**, a novel and efficient instruction-tuning data selection framework for Long-CoT reasoning. We leverage a LLM-as-a-Judge (Gu et al., 2024) to quantify instruction difficulty and prioritize more challenging problems. Additionally, we design an instruction-response joint ranker that combines rankings based on difficulty and trace length using a weighting factor. We conduct extensive experiments across three competition-level and six comprehensive mathematical benchmarks to validate the efficacy of our method. Built upon the OpenR1-Math dataset with 196K samples distilled from DeepSeek-R1, **SELECT2REASON** selects the top 10% instructions to fine-tune the Qwen2.5-Math-7B-Instruct model. Our method not only surpasses baselines but also matches or exceeds models trained on much larger datasets, such as the OpenR1-Qwen-7B (Face, 2025) with 94K samples and the DeepSeek-R1-Distill-Qwen-7B (Guo et al., 2025) with 800K samples, demonstrating its efficiency and effectiveness. Comprehensive ablation studies highlight the scalability of our approach under varying data sizes. Additionally, model fine-tuned on high-quality data selected by SELECT2REASON conducts more efficient exploration using fewer thinking tokens when generating solution with stronger performance. SELECT2REASON demonstrates strong generalization by enabling low-cost transfer to other long-CoT reasoning instruction pools like Chinese-DeepSeek-R1-Distill dataset (Liu et al., 2025) with 110K samples. Extensive case studies and visualizations support the effectiveness of our method.

Our contributions are summarized as follows: 1) We propose **SELECT2REASON**, a novel and efficient data selection framework for long-CoT instruction tuning. 2) We identify and validate key metrics—reasoning trace length and question difficulty—as strong heuristics for high-quality reasoning instruction selection. 3) We demonstrate state-of-the-art performance on multiple mathematical reasoning benchmarks using only a fraction of training data, with extensive experiments verifying scalability, robustness, and generalizability.

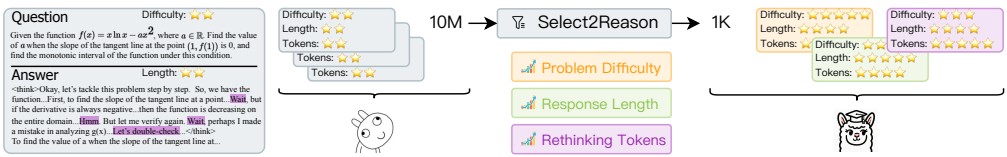

Figure 2: We select those data that can maximize reasoning ability via controlling the problem difficulty, response length, and the frequency of rethinking tokens.

## 2 RELATED WORK

**Reasoning of Large Language Models.** LLMs demonstrate notable chain-of-thought (CoT) (Wei et al., 2022) reasoning capabilities that are critical for addressing complex tasks such as mathematical problem solving, coding, and logical inference (Huang & Chang, 2022; Chen et al., 2025a; Li et al., 2025c). Various efforts aim to enhance LLMs' reasoning through different training stages. Previous works (Roziere et al., 2023; Shao et al., 2024) reinforce models to memorize reasoning patterns by injecting high-quality knowledge and rationales during pre-training. Furthermore, carefully curated datasets (Yu et al., 2023; Kim et al., 2023; Liu et al., 2023a) significantly boost complex reasoning performance through fine-tuning (Yuan et al., 2023). Some studies focus on scaling inference-time computation (Snell et al., 2024), such as employing self-consistency or reward-based verifiers to validate outcome or process on sampled candidate solutions (Wang et al., 2022; Lightman et al., 2023; Wang et al., 2023). Recently, researchers have observed planning and self-reflection behaviors in long-CoT responses of large reasoning models such as OpenAI-o1 (OpenAI, 2024), DeepSeek-R1 (Guo et al., 2025), Kimi-1.5 (Team et al., 2025), QwQ (Qwen Team, 2025) and Gemini Thinking (DeepMind, 2025), symbolizing a major breakthrough in complex reasoning. Open community projects (Face, 2025; Team, 2025) contribute by organizing synthetic datasets and distilling reasoning abilities from DeepSeek-R1 into smaller LLMs.

**Instruction-Tuning Data Selection.** Instruction-tuning data selection aims to identify high-utility subsets from large instruction pools to improve model performance and alignment. Early efforts emphasized human expert curation (Zhou et al., 2023), while recent work has explored automated selection using various metrics. GPT-based judgments of instruction-response quality are commonly used (Chen et al., 2023; Bukharin & Zhao, 2023; Liu et al., 2024; Zhang et al., 2024c; Li et al., 2025b), often enhanced with diversity signals (Liu et al., 2023b; Lu et al., 2023; Song et al., 2024; Yang et al., 2025; Chen et al., 2025b). Several studies leverage model-internal features such as loss (Li et al., 2023a; Du et al., 2023; Li et al., 2023b; Zhang et al., 2024b), gradients (Xia et al., 2024; Pan et al., 2024; Zhang et al., 2024a), perplexity (Li et al., 2024b; Mekala et al., 2024), and linguistic features (Cao et al., 2023; Zhao et al., 2024) to assess sample utility. Techniques like weak-to-strong supervision (Yang et al., 2024c; Li et al., 2024b; Mekala et al., 2024) and expert preference-aligned scoring (Ge et al., 2024) further enrich the selection space. With the advent of large reasoning models, LIMO (Ye et al., 2025), LIMR (Li et al., 2025a) and s1 (Muennighoff et al., 2025) observe that training with a few carefully crafted reasoning examples can achieve remarkable performance, highlighting the necessity of efficient minimal supervision. However, automatic selection of long-CoT reasoning instructions remains unexplored, where criteria have not been designed and verified.

## 3 PRELIMINARY EXPLORATION

In this section, we examine several metrics that may influence the frequency of rethinking tokens and conduct preliminary experiments to assess whether the metrics correlate with performance improvements, providing insights for selecting high-quality long-CoT instructions.

**Reasoning Traces with Varying Length in Instructions.** Prior work (Zhao et al., 2024) has shown that selecting instructions with the longest responses serves as a simple but tough-to-beat baseline. Recently, s1 (Muennighoff et al., 2025) employ an empirical study under 1K data budget to benchmark instruction subsets with longest length of response. We present the first systematic evaluation of how the length of the reasoning trace impacts instruction selection efficacy. Specifically, we sort the full instruction set $\mathcal{D}_p$ by the length of the reasoning trace $r$, and construct subsets $\mathcal{D}_L^k$, $\mathcal{D}_M^k$, and $\mathcal{D}_S^k$ corresponding to the top-$k\%$ longest, middle, and shortest traces, respectively, for $k \in \{2, 5, 10\}$.

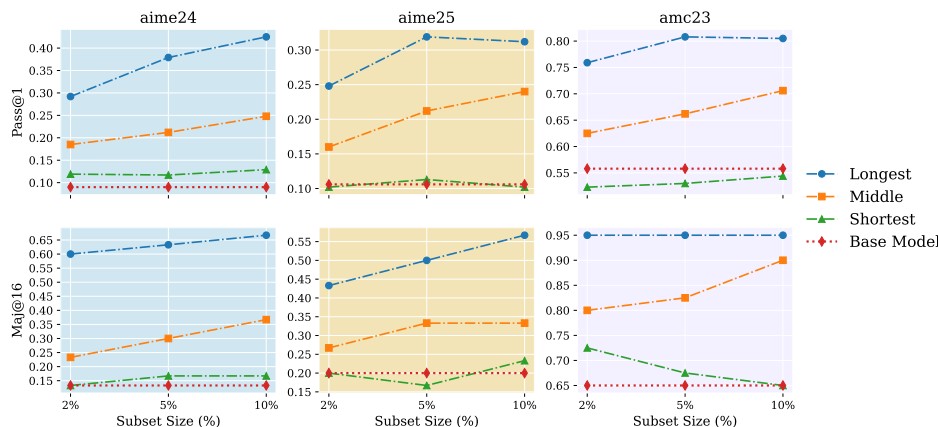

Figure 3: Performance across three expert-level benchmarks, using instruction subsets selected based on the length of reasoning traces: the longest, the shortest, and the middle.

As shown in Figure 3, models fine-tuned on $\mathcal{D}_L^k$ consistently outperform those trained on $\mathcal{D}_M^k$ and $\mathcal{D}_S^k$ across different dataset sizes, measured by metrics such as Pass@1 and Maj@16. Notably, $\mathcal{D}_M^k$ also yields positive gains over $\mathcal{D}_S^k$, highlighting a strong correlation between trace length and model improvement. While both $\mathcal{D}_L^k$ and $\mathcal{D}_M^k$ demonstrate scalable benefits with increasing subset size, the performance of model trained on $\mathcal{D}_S^k$ remains marginal or even negative—offering little to no improvement over the base model on AIME 24, and causing performance degradation on AIME 25 and AMC 23. This indicates that not only do short reasoning traces fail to activate the model's long-CoT reasoning capabilities, but they may also degrade its overall performance. Examples with different trace length are illustrated in Figure 4. Long reasoning traces incorporate more rethinking behaviors such as reflection, backtracking, and planning, and serve as higher-quality supervision signals. In contrast, short traces often omit substantive decision-making steps and, in some cases, explicitly bypass reasoning by using empty constructs like `<think>\n</think>`, rendering them ineffective. Statistics from Figure 1 further confirm this point: longer reasoning traces exhibit a higher frequency of reflective steps that begin with patterns such as *Wait*, *Alternatively* or *Maybe*. Motivated by these findings, we adopt **the longest reasoning traces as a simple, effective, and low-cost heuristic for data selection**, thereby avoiding the overhead of the reliance on costly human expert annotations (Zhou et al., 2023; Ye et al., 2025).

**Difficulty of Question.** Difficulty as a criterion for instruction selection is acknowledged across both alignment (Li et al., 2023a; 2024b; Mekala et al., 2024) and long-CoT reasoning (Muennighoff et al., 2025; Ye et al., 2025), with the prevailing intuition being that more challenging questions offer higher learning value. Same as trace length in Figure 1, instruction with harder question contains more rethinking tokens in reasoning trace. We validate this assumption through a straightforward empirical

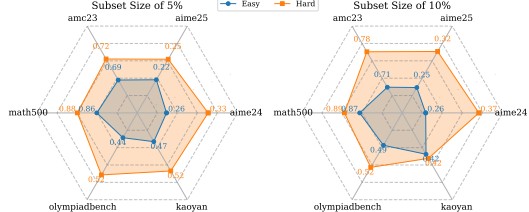

Figure 5: Pass@1 across six benchmarks. Easy and hard examples are selected separately.

study. Specifically, we perform short-CoT inference using the base model over a sampled subset of training instructions, and label instances as easy or hard based on whether the model successfully solves the question. This yields two subsets, $\mathcal{D}_E^k$ (easy) and $\mathcal{D}_H^k$ (hard), for $k \in \{5, 10\}$. As shown in Figure 5, models fine-tuned on $\mathcal{D}_H^k$ consistently outperform those trained on $\mathcal{D}_E^k$ across six mathematical benchmarks in two data scales. This aligns with prior findings and our hypothesis that instructions with harder question facilitate stronger supervision with richer training signals. Existing work (Ye et al., 2025; Muennighoff et al., 2025) has not established quantitative criteria for identifying the difficulty of questions, thereby limiting the prioritization of more challenging problems and **leaving the challenge of automated, difficulty-aware** instruction selection unaddressed.



## Length of Reasoning Traces

### Long Reasoning Trace

**Question:** Given the function $f(x) = x \ln x - ax^2$, where $a \in \mathbb{R}$. Find the value of $a$ when the slope of the tangent line at the point $(1, f(1))$ is 0, and find the monotonic interval of the function under this condition.

**Response:**
*<think>*

**[Problem Definition.]**
Okay, let's tackle this problem step by step. So, we have the function $f(x) = x \ln x - ax^2$...

**[Task Decomposition and Planning.]**
...First, to find the slope of the tangent line at a point, we need the derivative of $f(x)$... Then, the derivative of $-ax^2$ is $-2ax$...

**[Self Reflection.]**
...Wait, but if the derivative is always negative except at $x = 1$ where it's zero, then the function is decreasing on the entire domain $(0, \infty)$. Is that possible?...

**[Backtracking.]**
...Hmm. But let me verify again. Wait, perhaps I made a mistake in analyzing $g(x)$. Let's double-check...
*</think>*

**[Final Answer.]**
1. To find the value of $a$ when the slope of the tangent line at...

### Short Reasoning Trace

**Question:** Calculate the sum of the numbers 6 and 4. Add the number 30 to this sum.
**Response:**
*<think>*

**[No Self Reflection.]**
...Let me do that step by step. 6 plus 4... that should be 10, right? Yeah, 6 plus 4 equals 10.

**[No Backtracking.]**
...So putting it all together: first sum 6 and 4 to get 10, then add 30 to get 40. I don't think I made any mistakes here.
*</think>*

**[Final Answer.]**
The sum of 6 and 4 is calculated first: $6 + 4 = 10$. Next, add 30 to ...

### No Reasoning Trace

**Question:** $x, y \in \mathbb{R}^+, 3x + 2y = 12$, maximize $xy$
**Response:**
*<think>*
\n\n
**[Thinking Terminate.]**
*</think>*

**[Final Answer.]**
Given the constraint $3x + 2y = 12$, we aim to maximize...



Figure 4: Comparison of Instructions with Varying Reasoning Trace Lengths. Long reasoning trajectories incorporate more human-like cognitive behaviors—such as reflection, backtracking, and planning—and serve as higher-quality supervision signals during fine-tuning. In contrast, short traces often omit substantive decision-making steps and, in some cases, explicitly bypass reasoning by using empty constructs like `<think>\n</think>`, rendering them ineffective.

**Diversity.** Diversity has long been a central theme in instruction selection (Liu et al., 2023b; Lu et al., 2023; Yang et al., 2025; Ge et al., 2024), and its relevance to long-CoT reasoning has also been explored (Zhou et al., 2023; Muennighoff et al., 2025), particularly in the context of covering a broad range of mathematical domains and concepts. However, simple heuristics such as uniformly sampling from each domain offers no clear advantage over random selection for long-CoT reasoning instructions (Muennighoff et al., 2025). To further examine the role of diversity, we leverage *metadata* from the Open-R1-Math instruction set where problems are categorized into topics. We sample a domain-balanced subset $\mathcal{D}_D^k$ and compare it against a randomly sampled baseline subset $\mathcal{D}_R^k$ of the same size. As shown in Figure 11, the model fine-tuned on $\mathcal{D}_D^k$ does not exhibit significant performance gains over the baseline, and in some cases—such as Maj@16 on AMC 23, the performance curves nearly overlap. These results suggest that diversity may not contribute meaningfully for instruction selection in long-CoT reasoning, serving as a baseline only.

## 4 PROBLEM DEFINITION

**Long-CoT Reasoning.** We focus on the capability of large reasoning models (LRMs) to generate long chain-of-thought (CoT) reasoning traces for solving questions with verifiable answers. Given a question $q \in \mathcal{Q}$ and a model $M$ parameterized by $\theta$, the model is expected to generate a reasoning trace $r$ including steps $\{s_1, s_2, \ldots, s_n\}$, typically wrapped with `<think>` tokens, followed by a final answer $a \in \mathcal{A}$. Formally, the model outputs a pair $(r, a) \in \mathcal{R} \times \mathcal{A}$ such that:

$$f_M(q) = (r, a), \quad r = \texttt{<think>} \, s_1, s_2, \ldots, s_n \, \texttt{</think>}. \tag{1}$$

The quality of CoT reasoning trace is often characterized by the emergence of human-like behaviors such as planning, verification, reflection, and backtracking. High-quality reasoning traces exhibit these traits to navigate complex problem spaces and are more likely to converge on correct solutions.

**Instruction Selection.**    Instruction-tuning data selection aims to identify a optimal subset of reasoning instructions from a large instruction pool to enhance fine-tuning effectiveness. Given a reasoning instruction dataset $\mathcal{D}_p = \{I_i\}_{i=1}^N$, where each instruction $I_i = (q_i, r_i\|a_i)$ includes a question, a reasoning trace, and a final answer, and a proposed evaluation metric suite $\boldsymbol{\pi} = \{\pi_1, \pi_2, \ldots, \pi_k\}$ (e.g., quality, difficulty), our objective is to select a subset $\mathcal{D}_s \subseteq \mathcal{D}_p$ of size at most $K$ such that each selected instruction ranks among the top-$K$ under the metrics:

$$\mathcal{D}_s = \left\{ I \in Top_{\boldsymbol{\pi}}^K(\mathcal{D}_p) \right\}. \tag{2}$$

The supervised fine-tuning(SFT) objective is performed on $\mathcal{D}_s$ to update the model parameters $\theta$, thus minimizing the following negative log likelihood loss:

$$\min_\theta \ \mathcal{L}(\theta, \mathcal{D}_s) = -\frac{1}{|\mathcal{D}_s|} \sum_{(q,r,a)\in\mathcal{D}_s} \log p_\theta(r, a \mid q). \tag{3}$$

# 5    SELECT2REASON

We propose SELECT2REASON, an efficient instruction-tuning data selection method for long-CoT reasoning. Specifically, we leverage LLM-as-a-Judge to quantify question difficulty and propose a joint ranking strategy to balance difficulty with reasoning trace length.

**Quantifying Question Difficulty.**    To measure the difficulty of each instruction in the pool $\mathcal{D}_p$, prior methods (Li et al., 2023a; Mekala et al., 2024; Li et al., 2024b) often rely on model-specific loss or perplexity metrics, which are computationally expensive. We adopt an efficient alternative by using an LLM-as-a-Judge $M_j$ to quantifying difficulty scores. For each $q_i$, we prompt the model with *Please judge the difficulty of this instruction and return 1 if difficult or 0 if not.* The model outputs a probability distribution over the tokens *1* and *0*, from which we derive a scalar difficulty score:

$$difficulty(q_i) = \frac{e^{\log p(1|q_i)}}{e^{\log p(1|q_i)} + e^{\log p(0|q_i)}}. \tag{4}$$

To improve the adaptation of $M_j$ to this classification task, a small set $\mathcal{C}_j = \{(q, y)\}$ is designed for supervised fine-tuning, where $q \in \mathcal{D}_{Easy}^k$ is labeled *0* and $q \in \mathcal{D}_{Hard}^k$ is labeled *1* by model $M$ through whether the question can be directly solved referring to Section 3, and parameters $\theta_j$ are updated by minimizing the following negative log likelihood loss:

$$\min_{\theta_j} \ \mathcal{L}(\theta_j, \mathcal{C}_j) = -\frac{1}{|\mathcal{C}_j|} \sum_{(q,y)\in\mathcal{C}_j} \log p_{\theta_j}(y \mid q) \tag{5}$$

**Question-Response Joint Ranker.**    While we now have an efficient method to score questions via difficulty and responses via reasoning trace length, combining them in a principled manner remains a challenge. Inspired by prior work on multi-criteria ranking (Cao et al., 2023; Bukharin & Zhao, 2023), we aggregate rankings using a weighted scheme. Let $rank_d(I_i)$ and $rank_l(I_i)$ denote the rankings of instruction $I_i$ by question difficulty and reasoning trace length, we define the joint ranking as:

$$joint\_rank(I_i) = w \cdot rank_d(I_i) + (1 - w) \cdot rank_l(I_i), \tag{6}$$

where a weighting factor $w \in [0, 1]$ controls the trade-off between rankings by difficulty and trace length. The final selected subset by out methods for SFT is then:

$$\mathcal{D}_{\text{SELECT2REASON}} = \left\{ I \in Top_{joint\_rank}^K(\mathcal{D}_p) \right\}. \tag{7}$$

Table 1: Comparison between SELECT2REASON and baselines on the *OpenR1-Math-220k* pool through evaluation across nine benchmarks using Pass@1 and Maj@16 as metrics. We incorporate two models from open-source community for reference.

| Target Model | Data | AIME 24 | | AIME 25 | | AMC 23 | | MATH | Olympiad | Kaoyan | GK 23 | GK-Math | GK 24 |
|---|---|---|---|---|---|---|---|---|---|---|---|---|---|
| QWEN2.5-MATH-7B | Size | P@1 | M@16 | P@1 | M@16 | P@1 | M@16 | P@1 | P@1 | P@1 | P@1 | P@1 | P@1 |
| BASE MODEL | - | 0.090 | 0.133 | 0.106 | 0.200 | 0.558 | 0.650 | 0.842 | 0.394 | 0.472 | 0.649 | 0.781 | 0.637 |
| R1-DISTILL-QWEN | 800k | 0.544 | 0.833 | 0.417 | 0.600 | 0.895 | 0.950 | 0.896 | 0.551 | 0.618 | 0.810 | 0.880 | 0.692 |
| OPENR1-QWEN | 94k | 0.460 | 0.700 | 0.317 | 0.467 | 0.823 | 0.950 | 0.906 | 0.526 | 0.492 | 0.795 | 0.843 | 0.714 |
| FULL-POOL | 196k | 0.465 | 0.700 | 0.352 | 0.600 | 0.816 | 0.950 | 0.894 | 0.560 | 0.382 | 0.800 | 0.783 | 0.615 |
| RANDOM | | 0.331 | 0.600 | 0.267 | 0.367 | 0.753 | **0.950** | 0.878 | 0.510 | 0.467 | 0.740 | 0.789 | 0.626 |
| DIVERSE | | 0.327 | **0.667** | 0.267 | 0.433 | 0.750 | **0.950** | 0.846 | 0.493 | 0.467 | 0.745 | 0.809 | 0.659 |
| LONGEST | 10% | 0.425 | **0.667** | 0.312 | **0.567** | 0.805 | **0.950** | 0.898 | 0.535 | 0.548 | 0.795 | **0.892** | **0.747** |
| DIFFICULT | | 0.410 | 0.633 | 0.312 | 0.433 | 0.787 | 0.925 | 0.886 | 0.530 | 0.533 | 0.787 | 0.866 | 0.703 |
| SELECT2REASON | | **0.433** | **0.667** | **0.335** | **0.567** | **0.808** | **0.950** | **0.914** | **0.548** | **0.573** | **0.800** | **0.892** | 0.736 |

# 6 EXPERIMENT RESULTS AND ANALYSIS

**Datasets and Experiment Settings.** We adopt **OpenR1-Math-220k** (Face, 2025) as the data pool, which is a large-scale instruction set for long-CoT reasoning distilled from DeepSeek-R1 (Guo et al., 2025). We retain part of them that lead to a correct answer about **196k**. We employ **Qwen2.5-Math-7B-Instruct** (Yang et al., 2024b) as the backbone model. We adopt nine mathematics benchmark spanning multiple dimensions for evaluation, including three competition-level benchmarks which are AIME in 2024 & 2025, and AMC in 2023, and six comprehensive benchmarks such as MATH-500 (Hendrycks et al., 2021) and OlympiadBench (He et al., 2024) for math reasoning, and GAOKAO in 2023 & 2024 (Yang et al., 2024b), GAOKAO MATH Yang et al. (2024b) and KAOYAN (Ye et al., 2025) in Chinese math. More details are provided in Appendix A.1 and A.2.

**Main Results.** Table 1 presents the performance of SELECT2REASON across three competition-level and six comprehensive mathematics benchmarks, using Pass@1 (P@1) and Maj@16 (M@16) as evaluation metrics. We compare against DeepSeek-R1-Distill-Qwen-7B (Guo et al., 2025), which is trained on an unreleased set of 800k instructions, and OpenR1-Qwen-7B (Face, 2025), which uses 94k instructions from the OpenR1-Math dataset. We evaluate four baseline selection strategies: *Random*, which samples instructions uniformly from the pool; *Diverse*, which performs clustering and balanced sampling based on category metadata from OpenR1-Math; *Longest*, which selects instructions with the longest reasoning traces; and *Difficult*, which chooses top-ranked samples according to a difficulty quantifier. SELECT2REASON outperforms all baselines on most datasets, consistently achieving higher Pass@1 scores on competition-level benchmarks and matching the strongest baselines on the Maj@16 metric. Furthermore, the model trained on the subset filtered by SELECT2REASON surpasses both *Full-pool* models and open-source models on MATH-500. Notably, our method also maintains a strong lead over *Full-pool* training on nearly all comprehensive math benchmarks, with only a slight performance drop against the *Full-pool* on OlympiadBench. This may reflect a limitation in the generalization ability of *Full-pool*, whereas fine-tuning with a smaller, high-quality subset yields superior performance on Chinese benchmarks.

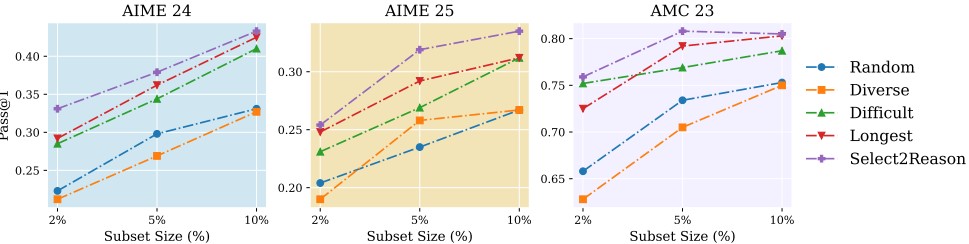

Figure 6: Performance across three benchmarks of baselines and our method in varying subset size.

**Performance of SELECT2REASON under different hyperparameter settings.** Figure 6 presents a statistical analysis of performance variation for both the baselines and SELECT2REASON across different subset sizes (2%, 5%, and 10%) on three datasets using Pass@1. SE-LECT2REASON consistently maintains a leading advantage. Moreover, as the subset size increases, the performance of SELECT2REASON generally improves in a stable manner.

Another critical hyperparameter is the weighted factor $w \in [0, 1]$ used in the joint ranker. Figure 7 shows this sensitivity analysis. When $w = 0$, the joint ranker degenerates to the length-based ranker; when $w = 1$, it becomes equivalent to the difficulty-based ranker. The best performance is achieved at $w = 0.25$, where the model fine-tuned on the top 10% subset reaches highest accuracy, as reported in Table 1. This indicates that the joint ranker achieves an effective balance in controlling the trade-off.

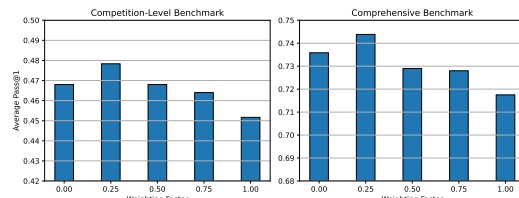

Figure 7: Average Pass@1 by adjusting the weighting factor of joint ranker in SELECT2REASON.

**SELECT2REASON improves long-CoT reasoning efficiency by sampling high-quality data.** The relationship between performance and output tokens on AIME 25 is illustrated in Figure 8. In contrast to the increasing response lengths observed during conducting pure RL on pre-trained models (Guo et al., 2025), SFT exhibits a different distribution: models with stronger performance tend to generate shorter outputs. **This suggests that when long-CoT reasoning is effectively activated via SFT, models can produce more efficient exploratory solutions.** Further statistical analysis is presented in Figure 12, which shows the frequency of rethinking tokens used by fine-tuned models on AIME 25. The model trained on subsets selected by SELECT2REASON consistently uses fewer rethinking tokens across all data sizes, supporting our hypothesis that it enables more efficient reasoning. A case study is provided in Figure 10, where an LRM fine-tuned on limited and low-quality instructions attempts to use a large number of rethinking tokens during inference, but exhibits limited effective reflection. In comparison, a model trained on higher-quality instructions corrects its reasoning path and reaches the correct solution with fewer rethinking steps.

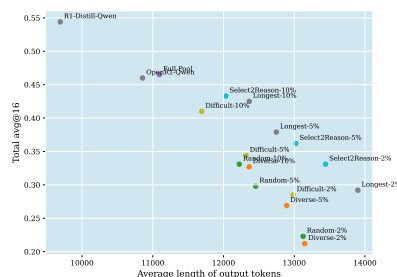

Figure 8: Relationship between performance on AIME 25 and output length.

**SELECT2REASON demonstrates strong generalization capabilities by enabling low-cost transfer to other Long-CoT reasoning instruction pools.** To assess the generalizability of SE-LECT2REASON on *Chinese-DeepSeek-R1-Distill-data*, we directly apply the joint ranker trained on OpenR1-Math-220k. Results in Table 2 show that fine-tuning model on only the top 10% subset selected by SELECT2REASON outperforms baselines. Notably, since this data pool contains a large proportion of generic, non-reasoning instructions, we conclude that this dilutes the model's ability to acquire strong reasoning capabilities. Case studies of joint ranking are presented in Appendix A.5, despite not being trained on this specific instruction pool, the joint ranker still successfully identifies high-quality reasoning instructions, demonstrating the notable generalizability of SELECT2REASON.

Table 2: Generalizability of SELECT2REASON on the *Chinese-DeepSeek-R1-Distill-data* pool.

| Target Model | Data | AIME 24 | | AIME 25 | | AMC 23 | | MATH | Olympiad | Kaoyan | GK 23 | GK-Math | GK 24 |
|---|---|---|---|---|---|---|---|---|---|---|---|---|---|
| QWEN2.5-MATH-7B | Size | P@1 | M@16 | P@1 | M@16 | P@1 | M@16 | P@1 | P@1 | P@1 | P@1 | P@1 | P@1 |
| BASE MODEL | - | 0.090 | 0.133 | 0.106 | 0.200 | 0.558 | 0.650 | 0.842 | 0.394 | 0.472 | 0.649 | 0.781 | 0.637 |
| FULL-POOL | 110K | 0.181 | 0.267 | 0.158 | 0.300 | 0.633 | 0.800 | 0.798 | 0.367 | 0.412 | 0.668 | 0.718 | 0.626 |
| RANDOM | | 0.181 | 0.200 | 0.140 | 0.167 | 0.620 | 0.725 | 0.840 | 0.431 | 0.372 | **0.701** | 0.775 | 0.626 |
| DIVERSE | | 0.176 | 0.233 | 0.144 | 0.233 | 0.618 | 0.750 | 0.822 | 0.416 | 0.377 | 0.692 | 0.770 | 0.641 |
| LONGEST | 10% | 0.221 | 0.367 | 0.173 | 0.233 | 0.656 | **0.850** | 0.846 | **0.459** | 0.457 | 0.688 | 0.821 | **0.703** |
| DIFFICULT | | **0.258** | **0.400** | 0.194 | 0.267 | 0.627 | 0.800 | 0.848 | 0.412 | **0.462** | 0.691 | 0.795 | 0.681 |
| SELECT2REASON | | 0.242 | **0.400** | **0.206** | **0.367** | **0.689** | 0.825 | **0.860** | 0.450 | **0.462** | 0.699 | **0.840** | **0.703** |

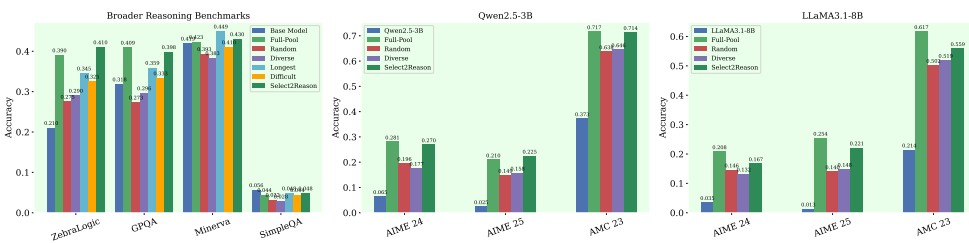

Figure 9: (a) Generalization in broader reasoning tasks. (b)(c) Generalization across various LLMs.

**SELECT2REASON demonstrates robust generalization across domains and model scales.** Beyond the mathematical domain, we extend our evaluation to broader reasoning tasks, including logical inference, scientific QA, and commonsense reasoning. As summarized in Figure 9 (a), SELECT2REASON consistently achieves superior performance compared to baseline strategies, particularly on benchmarks such as ZebraLogic and GPQA, where long-CoT reasoning is essential. Furthermore, to assess robustness across model families and scales, we fine-tune two additional open-source models, Qwen2.5-3B-Instruct (Yang et al., 2024a) and LLaMA-3.1-8B-Instruct (Dubey et al., 2024). The results in Figure 9 (b)(c) confirm that SELECT2REASON maintains its advantage across both smaller-scale models and different architectures. Specifically, while absolute performance decreases with model size, the relative improvements over baselines remain consistent, validating that the effectiveness of our method is not confined to a single model family or scale.

**The data picked by SELECT2REASON yields higher quality comparing with datasets in prior methods.** We conduct a comparative analysis with LIMO (Ye et al., 2025) and S1 (Muennighoff et al., 2025) in data quality. Specifically, we select approximately 1k long-CoT instructions from each synthesized data pool to evaluate performance across five benchmarks. As shown in Table 3, the model trained on instructions selected by SELECT2REASON consistently outperforms those trained on data selected by LIMO and S1.1. Furthermore, we

Table 3: Comparison of selection with prior ways.

| Model | Data Size | AIME 24 | AIME 25 | AMC 23 |
|---|---|---|---|---|
| BASE MODEL | - | 0.090 | 0.106 | 0.558 |
| *Individual Corpus* | | | | |
| Qwen2.5-S2R | 982 | **0.283** | **0.237** | **0.728** |
| Qwen2.5-S1.1 | 1k | 0.225 | 0.198 | 0.669 |
| Qwen2.5-LIMO | 871 | 0.206 | 0.210 | 0.627 |
| *Full Corpus of S1* | | | | |
| Qwen2.5-S1.0 (FULL) | 59k | 0.224 | 0.169 | 0.588 |
| Qwen2.5-S1.0 (1K) | 1k | 0.202 | 0.154 | **0.614** |
| Qwen2.5-S1.0-S2R | 1k | **0.238** | **0.177** | 0.606 |

apply SELECT2REASON to the full instruction pool used by S1.0, which comprises 59k examples with Gemini Flash Thinking responses (the full pool used by LIMO is not publicly available) to select a 1k subset, and again observe improved performance over the original selected 1k subset of S1.0.

**SELECT2REASON achieves significant training efficiency with minimal selection overhead.** We conduct a detailed cost-benefit analysis of SELECT2REASON to assess its computational efficiency. As shown in Table 4, the total overhead introduced by the selection process is minimal compared to the cost of training on the full instruction pool. Notably, this results in a 75% reduction in training time without compromising performance. Additionally, when applied to a new data pool, the judge model generalizes effectively without retraining, and the inference stage completes within 3 minutes. This demonstrates the transferability and amortized cost of the pipeline.

## 7 CONCLUSION

In summary, while recent large reasoning models exhibit remarkable long-CoT reasoning abilities, effective instruction selection remains an underexplored challenge. Our study identifies reasoning trace length and problem difficulty as strong, quantifiable heuristics for high-quality data selection. Building on these insights, we introduce SELECT2REASON, an novel and efficient instruction-tuning data selection framework for long-CoT reasoning. Extensive empirical validation demonstrates that models trained on our selected subsets achieve superior reasoning performance using significantly less data, paving the way for cost-effective and high-quality instruction tuning in long-CoT tasks.

## ETHICS STATEMENT

We conduct our experiments on publicly available, open-source datasets that are curated and maintained by community contributors. While we have made best efforts to perform manual inspection and filtering, it is possible that a small fraction of the data may still contain issues such as fairness concerns, biases, or inadvertent privacy leaks. Furthermore, once our proposed SELECT2REASON framework is released as open source, we cannot fully prevent its application on datasets that may involve ethical risks. To mitigate these concerns, we will strive to provide clear documentation and usage guidelines with the release, encourage responsible adoption within the community, and actively call for further research on automated auditing techniques to detect and address ethical issues in large-scale instruction datasets.

## REPRODUCIBILITY STATEMENT

We have provided detailed descriptions of our implementation in the **Experiment Settings** section, including preprocessing procedures, dataset and model selection, experimental hyperparameters, and the computing environment. Due to the large scale of the datasets and the need to preserve anonymity during the double-blind review process, we do not release code or processed datasets at this stage. However, we commit to releasing executable code and the processed datasets after the review process is completed, ensuring full reproducibility of our results.

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

## A APPENDIX

### A.1 EXPERIMENTAL SETTINGS

We adopt **OpenR1-Math-220k** (Face, 2025) as the data pool, which is a large-scale instruction set for long-CoT reasoning distilled from DeepSeek-R1 (Guo et al., 2025) using math problems from NuminaMath (Li et al., 2024a). We retain part of them that lead to a correct answer about **196k**. We also adopt **Chinese-DeepSeek-R1-Distill-data** (Liu et al., 2025), a open-source dataset containing 110k Chinese instructions spanning mathematics, STEM, and general domains, with Long-CoT responses generated by DeepSeek-R1 for validating generalization. We employ **Qwen2.5-Math-7B-Instruct** (Yang et al., 2024b) as the backbone model to perform full parameters supervised fine-tuning on selected instruction subsets. We extend the model's context length from 4,096 to 16,384 via RoPE (Su et al., 2024) scaling, increasing the RoPE frequency from 10k to 300k. We conduct experiments on a Linux server equipped with 8 A100-SXM4-40GB GPUs. Utilizing the LLaMA-Factory framework (Zheng et al., 2024), we set the sequence limit of 16,384, batch size to 1, gradient accumulation steps to 4, and learning rate to 5e-5 with a warmup ratio of 0.1, followed by a cosine decay schedule towards zero. The training epochs is 3 for any size of subset. For the judge model, we apply the LoRA technique (Hu et al., 2022), with the rank of 16, alpha of 32, and dropout rate of 0.1, training for 1 epoch. We utilize DeepSpeed ZeRO-3 (Rajbhandari et al., 2020) and FlashAttention2 (Dao, 2023) to accelerate computations on GPUs.

### A.2 EVALUATIONS SETTINGS

We adopt nine mathematics benchmark spanning multiple dimensions for evaluation, including three competition-level benchmarks which are AIME in 2024 & 2025, and AMC in 2023, and six comprehensive benchmarks such as MATH-500 (Hendrycks et al., 2021) and OlympiadBench (He et al., 2024) for math reasoning, and GAOKAO in 2023 & 2024 (Yang et al., 2024b), GAOKAO MATH Yang et al. (2024b) and KAOYAN (Ye et al., 2025) to validate the generalization capability in Chinese math. For broader reasoning tasks, we include GPQA (Rein et al., 2024), Minerva (Lewkowycz et al., 2022), ZebraLogic (Lin et al., 2025) and SimpleQA (Wei et al., 2024). Following (Yang et al., 2024b; Guo et al., 2025), the system prompt for evaluation is *Please reason step by step, and put your final answer within \boxed{}*. For three competition-level mathematical benchmarks, 16 solutions per problem are sampled to mitigate fluctuations due to the limited number of questions, setting temperature to 0.6 and top_p to 0.95. For benchmarks like challenging mathematics and Chinese mathematics, we employ greedy decoding by setting temperature to 0 and top_p to 1, sampling only one solution per problem to compute accuracy. Correctness is verified using a rule-based evaluation framework (Ye et al., 2025), assessing numerical and formulaic accuracy. Scores are calculated referring to (Yang et al., 2024b), yielding metrics including pass rate (Pass@1) and majority voting accuracy (Maj@16). We use the vLLM inference engine (Kwon et al., 2023) to achieve inference acceleration, where sequence is limited to 16,384.

## A.3 CASE STUDY OF EFFICIENT EXPLORATION IN REASONING

---

**Case Study of Efficient Exploration in Reasoning**

**Question:**

Suppose $\triangle ABC$ has angles $\angle BAC = 84^\circ$, $\angle ABC = 60^\circ$, and $\angle ACB = 36^\circ$. Let $D, E,$ and $F$ be the midpoints of sides $\overline{BC}$, $\overline{AC}$, and $\overline{AB}$, respectively. The circumcircle of $\triangle DEF$ intersects $\overline{BD}$, $\overline{AE}$, and $\overline{AF}$ at points $G, H,$ and $J$, respectively. The points $G, D, E, H, J,$ and $F$ divide the circumcircle of $\triangle DEF$ into six minor arcs, as shown. Find $\widehat{DE} + 2 \cdot \widehat{HJ} + 3 \cdot \widehat{FG}$, where the arcs are measured in degrees.

**Long-CoT Reasoning with Low-Efficiency:**

<think>

Okay, so I need to find the sum of some arcs on the circumcircle of triangle DEF. Let me start by understanding the problem step by step.

…

Alternatively, consider that 84° + 60° + 36° = 180°, but the combination DE + 2*HJ + 3*FG equals 312°, which is 84°*4 - 12°, **not helpful.**

Alternatively, maybe the answer is 3*(84° + 60° + 36°) - 120°, but that's 432 - 120 = 312, **again not helpful.**

Alternatively, perhaps the answer is 2*(180°) - 180° + 36°, which is 36°, **but that doesn't align.**

…

Alternatively, maybe I made a miscalculation. Let's check:

DE = 84°, HJ = 60°, FG = 36°.

DE + 2*HJ + 3*FG = 84 + 2*60 + 3*36 = 84 + 120 + 108 = 312°.

Yes, that's correct. However, 312° is equal to 360° - 48°, which might relate to the orthic triangle or something else, **but I don't see the connection**.

Alternatively, maybe the answer is 180°, considering that in some configurations, the sum of arcs can be 180°, **but I can't see how**.

…

Given that, after careful consideration and multiple approaches, I think the answer is 312°, but since this is a competition problem, it's likely that the answer is a nice number related to the original angles. However, given the combination 84 + 2*60 + 3*36, which yields 312, I think that's the correct answer. </think>

**Long-CoT Reasoning with High-Efficiency:**

<think>Okay, so I'm trying to solve this geometry problem about triangle ABC with given angles, midpoints, and intersections on its circumcircle.

…

Alternatively, perhaps recognizing that in triangle ABC, the centroid G divides the median in ratio 2:1, so the power of point G with respect to the nine-point circle might be useful.

Alternatively, perhaps using spiral similarity.

But perhaps stepping back, given the approximate measures, the total is around 334.5°, which is close to 360°.

But the problem expects an exact value, likely an integer degree measure.

Given the initial angles are multiples of 6°, the answer is likely multiple of 6°, perhaps 360° - something.

But wait, 334.5° is approximately 334.5°, which is 360 -25.5°, but not particularly meaningful.

Alternatively, maybe the exact values are 72°,24°,72°, leading to 72+48+216=336°, which is 336°, close to our approximate 334.5°, but not exact.

Alternatively, perhaps there's a miscalculation in the approximate measures.

*Wait, let me re-examine the central angles.*

*For arc DE, between D and E.*

Vectors v_D=(0.2498,0.3445), v_E=(-0.2502,0.3445).

…

If that's the case, then:

measure(arc DE)=2*angle at C=72°

measure(arc HJ)= angle at B - angle at C=24°

measure(arc FG)=2*angle at C=72°

Thus, the expression: measure(arc DE) + 2*measure(arc HJ) +3*measure(arc FG)=72° +2*24° +3*72°=72 +48 +216=336°

So, 336°, which is 360° -24°, which is angle at B.

So, maybe that's the exact value.

Thus, boxed, it's \boxed{336}

---

Figure 10: A case study of efficient exploration in reasoning on AIME 25. The emergence of more rethinking tokens in long-CoT reasoning does not lead to improved efficiency. In contrast, stronger LRMs reach the correct reasoning trace more efficiently.

## A.4 DIVERSITY OF INSTRUCTION

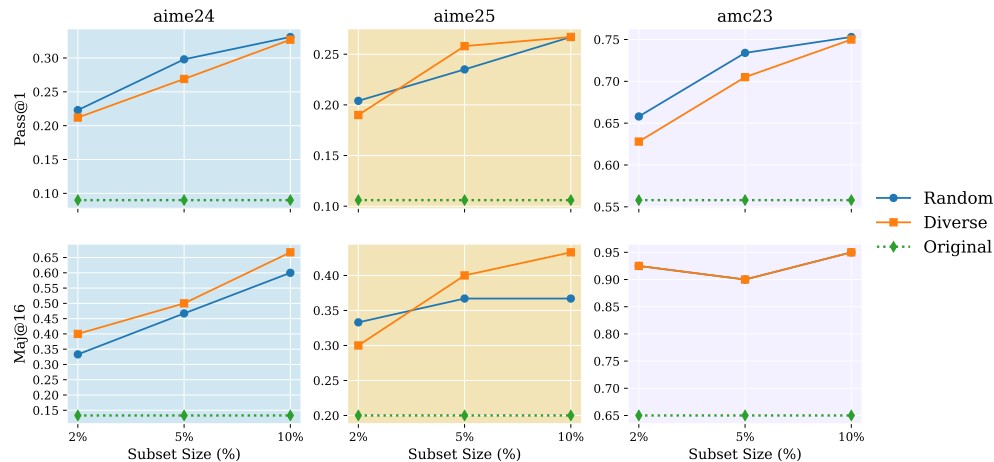

Figure 11: Performance across three expert-level benchmarks. Subset size refers to the proportion selected from data pool by length reasoning trace, either diverse or random.

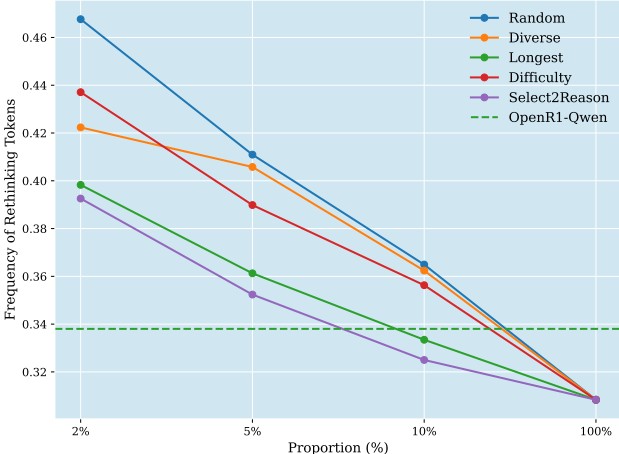

Figure 12: Relationship between performance on AIME 25 and output length.

Table 4: Computational overhead of data selection.

| Type | Data Size | Strategy | Time Cost | GPU Usage |
|---|---|---|---|---|
| **Open-R1-Math** | | | | |
| Full Pool | 196k | Full-SFT | 40 hours | 8×40GB |
| Judge Model | 4k | LoRA-SFT | 10 mins | 8×40GB |
| Difficulty Scoring | 196k | Inference | 11 mins | 2×40GB |
| Length Ranking | 196k | - | - | - |
| Select2Reason | 19.6k | Full-SFT | 10 hours | 8×40GB |
| **Chinese-R1-Distill** | | | | |
| Full Pool | 110k | Full-SFT | 10 hours | 8×40GB |
| Difficulty Scoring | 110k | Inference | 3 mins | 2×40GB |
| Length Ranking | 110k | - | - | - |
| Select2Reason | 11k | Full-SFT | 3 hours | 8×40GB |

## A.5 Case study of data selection by joint ranker in Chinese-DeepSeek-R1-Distill-data.

---

**Case study of joint ranking in Chinese-DeepSeek-R1-Distill-data**

**Instruction: (Difficulty Score: 0.7549)**

已知数列 $\left\{a_{n}\right\},\left\{b_{n}\right\}$ 均为各项都不相等的数列，$S_{n}$ 为 $\left\{a_{n}\right\}$ 的前 $n$ 项和，$a_{n+1} b_{n}=S_{n}+1\left(n \in N^{*}\right)$。(1) 若 $a_{1}=1, b_{n}=\frac{n}{2}$，求 $a_{4}$ 的值; (2)...(3)...

Given two sequences $\{a_n\}$ and $\{b_n\}$ where all terms are distinct, and $S_n$ is the sum of the first $n$ terms of $\{a_n\}$, with the relationship $a_{n+1} b_n = S_n + 1$ for all $n \in \mathbb{N^*}$. (1) If $a_1 = 1$ and $b_n = \frac{n}{2}$, find the value of $a_4$;(2)...(3)...

**Response: (Length of Reasoning Trace: 6427)**

\<think\>好，我现在要解决这三个题目，先从第一题开始吧。

Alright, I'll start solving these three problems, beginning with the first one.

...n=3时，S3=1+4+6=11，右边11+1=12，左边a4\*(3/2)=12 →a4=8，正确。没问题。所以（1）的结果是a4=8。接下来是题目（2）

...When n=3, $S_3 = 1 + 4 + 6 = 11$, the right-hand side is $11 + 1 = 12$, the left-hand side is $a_4 \cdot \frac{3}{2} = 12 \Rightarrow a_4 = 8$, correct. So the answer to (1) is $a_4 = 8$. Next, problem (2)...

...不过需要注意首项是否与n有关，显然这里的表达式对于每个n来说，除了q^n外，都是常数，所以确实构成等比数列。因此，题目（2）得证。

...Note that the first term's dependency on $n$ needs attention. Clearly, for each $n$, aside from $q^n$, the rest is constant, so the sequence is geometric. Thus, problem (2) is proven.

...然而，这样的方法可能需要较多的计算，或许有更简洁的方法。另一种思路是利用数列的递推关系。

...However, this method involves a lot of computation; perhaps a simpler way exists. Another approach is to use the recurrence relation.

...综上，三个问题的解答如上：（1）a4=8；（2）证明{b\_n +1/(1−q)}是公比为1/q的等比数列；（3）充要条件为d=1/2。

In summary, the solutions to the three problems are as follows: (1) $a_4 = 8$; (2) Prove that $\{b_n + \frac{1}{1 - q}\}$ is a geometric sequence with common ratio $1/q$; (3) The necessary and sufficient condition is $d = \frac{1}{2}$. \</think\>

(1) 由已知条件 $a_{n+1}b_n = S_n + 1$，...故充要条件为 $\boxed{d = \frac{1}{2}}$。

(1) From the given condition $a_{n+1}b_n = S_n + 1$, ... thus the necessary and sufficient condition is $\boxed{d = \frac{1}{2}}$.

---

**Instruction: (Difficulty Score: 0.1824)**

Nina想用她的零用钱买一个新的视频游戏。游戏的价格是50美元。Nina还了解到有10%的销售税。她每周收到10美元的零用钱，并认为她可以存一半。Nina需要多少周才能用她的储蓄买到新的视频游戏？

Nina wants to buy a new video game with her allowance. The game costs \$50. She also found out there's a 10% sales tax. She receives \$10 per week and plans to save half of it. How many weeks will it take for Nina to save enough to buy the game?

**Response: (Length of Reasoning Trace: 340)**

\<think\>嗯，好的，我现在得帮Nina算算她需要存多少周才能买到那个视频游戏。

Alright, let's calculate how many weeks Nina needs to save to buy that video game.

...接下来，她每周存多少钱。零用钱是10美元，存一半的话，就是10除以2，等于5美元每周。她每周存5美元，总需要的钱是55美元，所以需要多少周呢？

...Then, how much does she save weekly? She gets \$10 and saves half, so \$10 ÷ 2 = \$5 per week. The total needed is \$55, how many weeks is that?

...不过我得再检查一下步骤有没有错。首先计算总成本是否正确。

...But I should double-check the steps. First, let's verify if the total cost is calculated correctly. \</think\>

Nina需要计算购买含税后的游戏总成本，并确定需要多少周的储蓄才能达到这个金额。...最终，Nina需要 $\boxed{11}$ 周的储蓄才能购买游戏。

Nina needs to calculate the total cost including tax and determine how many weeks of savings are required to reach that amount. ... In the end, Nina needs $\boxed{11}$ weeks of saving to buy the game.

---

**Instruction: (Difficulty Score: 0.0420)**

绿豆稀饭的颜色是怎样的？

What is the color of mung bean porridge?

**Response: (Length of Reasoning Trace: 501)**

\<think\>嗯，用户问的是绿豆稀饭的颜色是怎样的。首先，我需要确认自己对绿豆稀饭的了解。

The user is asking what color mung bean porridge is. First, I need to confirm my understanding of it.

...还有可能加入其他食材，比如冰糖或百合，但用户的问题没有提到这些，所以暂时不考虑。

...It may also include other ingredients like rock sugar or lily bulbs, but since the user didn't mention them, we'll set them aside for now.

...总结一下，绿豆稀饭的颜色通常是浅黄绿色或浅褐色，具体取决于绿豆的烹饪程度、水质和烹饪方法

...To summarize, the color of mung bean porridge is usually light yellow-green or light brown, depending on how it's cooked, the water quality, and the method used. \</think\>

绿豆稀饭的颜色通常呈现为\*\*浅黄绿色或浅褐色\*\*，具体取决于以下几个因素：...若追求颜色鲜绿，可缩短熬煮时间或调节水的酸碱度。

The color of mung bean porridge is typically \*\*light yellow-green or light brown\*\*, depending on several factors: ... 4. \*\*Additives\*\*: Sugar or dried fruits can affect the final color. To retain a green hue, shorten cooking time or slightly acidify the water

---

Figure 13: Case study of data selection by joint ranker in Chinese-DeepSeek-R1-Distill-data.

## B    LIMITATIONS

Despite the promising results of SELECT2REASON in enhancing long-CoT reasoning capabilities, several limitations remain. First, due to constraints in computational resources and training costs, our experiments are primarily conducted on medium-scale models, and the scalability of our method to larger models remains to be explored. Second, the current study relies on existing instruction datasets, while automated instruction evolution strategies to improve data quality are yet to be developed. Finally, although our analysis reveals correlations between reasoning trace length, problem difficulty, and rethinking behaviors, the interpretability of how long-CoT capabilities are activated and how reflective reasoning emerges during SFT remains an open question for future work.

## C    BROADER IMPACT

Our work aims to improve the efficiency of instruction tuning for long-CoT reasoning by selecting high-quality data subsets. This approach significantly reduces the need for large-scale supervised fine-tuning on massive instruction datasets, thereby lowering computational cost, energy consumption, and dependency on high-end hardware. By enabling stronger performance using only a fraction of training data, SELECT2REASON contributes to the development of more sustainable and environmentally friendly AI systems.

## D    THE USE OF LARGE LANGUAGE MODELS (LLMS)

During the completion of this thesis, the scenarios involving the use of LLMs included: using code-completion tools to assist with experiments, and using ChatGPT to polish the draft after the initial writing was completed. LLMs were not involved in any aspects such as the development of research ideas, literature review, and so on.

