# OpenReview forum: "SELECT2REASON: Efficient Instruction-Tuning Data Selection for Long-CoT Reasoning"
_ICLR.cc/2026/Conference — ICLR 2026 Conference Withdrawn Submission_

### Official Review · Reviewer_eZ83 · 2025-10-31

**Soundness:** 2
**Presentation:** 2
**Contribution:** 1
**Rating:** 2
**Confidence:** 4

**Summary:**

The paper introduces a heuristic for data selection for optimal training on long-CoT data. It combines selection based on response length and difficulty, using LLM-as-a-Judge for the latter. The method ranks data samples based on these two metrics and selects the final subset via weighred rank aggregation.

The authors show that the method yields better performance when taking a 10% subset of OpenR1-Math-220k dataset, compared to data selection based solely on problem difficulty or response lengths, and random selection.

**Strengths:**

Presented method yields stronger performance than tested simpler heuristics, which might have practical applications for training on CoT data with limited compute budget.
The paper is well written, and method clearly presented and easy to understand.

**Weaknesses:**

The main issue of the paper might be the limited novelty. As mentioned by the authors (Introduction & Preliminary Exploration sections), data selection based on length and difficulty was already explored by multiple authors, and this work combines both metrics by weighted ranking.

The method is complex, and provides only marginal gains over much simpler heuristic based on generation length. Moreover, the LLM-as-a-Judge used for assesing difficulty of the problem relies heavily on the capability of the teacher-model to determine difficulty based on the problem definition only.

As for more actionable problems:
- The main results are shown for already capable Qwen2.5-Math-7B-Instruct model. Additional results for other models are missing baselines -- based on length and difficulty. How does the method compare to these baselines on different models?
- The authors showed that the number of "rethinking" tokens correlate with both, the problem difficulty and response length. That seems like a good baseline heuristic for data selection, as it would be proxy for both metrics. How does the method compare to this simple heuristic?
- Figures are hardly readable. It would be beneficial to align font size of figures with the one in the main text.
- The authors claim achieving state-of-the-art performance on multiple benchmarks. This is a false claim, as there are models with better performance, eg. Qwen3 model family. It should be corrected/clarified.

**Questions:**

As mentioned in weaknesses, here are the actionable problems:
- The main results are shown for already capable Qwen2.5-Math-7B-Instruct model. Additional results for other models are missing baselines -- based on length and difficulty. How does the method compare to these baselines on different models?
- The authors showed that the number of "rethinking" tokens correlate with both, the problem difficulty and response length. That seems like a good baseline heuristic for data selection, as it would be proxy for both metrics. How does the method compare to this simple heuristic?
- Figures are hardly readable. It would be beneficial to align font size of figures with the one in the main text.
- The authors claim achieving state-of-the-art performance on multiple benchmarks. This is a false claim, as there are models with better performance, eg. Qwen3 model family. It should be corrected/clarified.

---

> ### Author Response · Authors · 2025-11-19
> **Response to Reviewer eZ83**
>
> We thank the reviewer for the constructive feedback and appreciate the opportunity to clarify our contributions. Below we respond to each of the points raised.
>
> ---
>
> > **W1: Limited novelty**
>
> While trace length and difficulty have been discussed in prior studies, our perspective and task setting differ substantially from earlier work. Long-CoT reasoning introduces new challenges that do not appear in short-CoT or standard instruction-tuning scenarios. In long-chain settings, trace length reflects not verbosity but multi-step decomposition, reflective correction, and iterative reasoning, while difficulty interacts with long-CoT dynamics in more complex ways. As shown in our preliminary exploration and ablation studies, combining these two signals is not a trivial stacking of existing heuristics; rather, it is motivated by the structural characteristics of long-CoT instruction data and yields consistent performance gains across benchmarks and subsets.
>
> ---
>
> > **W2: Performance gains and reliability of the judge model**
>
> Regarding the performance gains of Select2Reason, the improvements are clearly supported by the results in Table 1 and Table 2. Whether the improvements appear “marginal” depends on the perspective; under the constrained setting of using only 10% of the data, even small absolute gains translate into meaningful efficiency improvements. Moreover, Figure 6 shows that the benefit of Select2Reason remains robust as subset size varies, indicating that the combination of length and difficulty forms a stable signal compared with simpler heuristics.
>
> On the reliability of the judge model, we clarify that our LLM-as-a-Judge only uses a **7B** model rather than a larger teacher model. Thus, the claim that our approach **heavily relies on a powerful teacher is a misunderstanding**.
>
> ---
>
> > **Q1: Comparison to length- and difficulty-based baselines on different models**
>
> Following the reviewer’s suggestion, we conducted additional experiments on Qwen2.5-3B and LLaMA3.1-8B. We evaluated length-based, difficulty-based, random, and diverse baselines using 10% sampled subsets. Results are as follows:
>
> **Qwen2.5-3B**
>
> | Method        | AIME24 | AIME25 | AMC23 |
> |---------------|--------|--------|-------|
> | Qwen2.5-3B    | 0.065  | 0.025  | 0.373 |
> | Full-Pool     | 0.281  | 0.210  | 0.717 |
> | Random        | 0.196  | 0.149  | 0.638 |
> | Diverse       | 0.177  | 0.158  | 0.646 |
> | Difficult     | 0.232  | 0.189  | 0.661 |
> | Longest       | 0.257  | 0.208  | 0.707 |
> | Select2Reason | 0.270  | 0.225  | 0.714 |
>
> **LLaMA3.1-8B**
>
> | Method        | AIME24 | AIME25 | AMC23 |
> |---------------|--------|--------|-------|
> | LLaMA3.1-8B   | 0.035  | 0.013  | 0.214 |
> | Full-Pool     | 0.208  | 0.254  | 0.617 |
> | Random        | 0.146  | 0.140  | 0.502 |
> | Diverse       | 0.132  | 0.148  | 0.519 |
> | Difficult     | 0.149  | 0.178  | 0.528 |
> | Longest       | 0.172  | 0.200  | 0.554 |
> | Select2Reason | 0.167  | 0.221  | 0.559 |
>
> ---
>
> > **Q2: Rethinking tokens as a heuristic for data selection**
>
> Util now we haven't use rethinking-token frequency as a selection metric. Keywords associated with reflective reasoning serve only as an explanatory or interpretive lens, aligned with observations in prior work. Such human-defined keywords cannot reliably capture the depth or emergence of genuine long-CoT reasoning, and their repetition can even introduce noise or redundancy. For these reasons, we instead rely on **trace length and difficulty**, two signals that are more stable, more generalizable, and less susceptible to keyword-level artifacts. Rethinking tokens are used solely to validate correlations and causal interpretations, not as direct selection indicators.
>
> ---
>
> > **Q3: Figures are hardly readable**
>
> We thank the reviewer for the suggestion. We will adjust and unify the font sizes for all figures and update them in the revised version to ensure clarity and readability.
>
> ---
>
> > **Q4: Claiming state-of-the-art performance**
>
> We acknowledge that our wording have caused confusion. Select2Reason achieves a better data selection performance relative to comparable baselines, rather than the highest raw benchmark score across all existing models. We will revise the phrasing accordingly and ensure that the comparison is presented accurately in the updated manuscript.
>
> ---
>
> Thank you again for your valuable comments. We will incorporate all improvements and clarifications into the revised version.

---

> > ### Comment · Reviewer_eZ83 · 2025-11-21
> >
> > Thank you for your response.
> >
> > >our perspective and task setting differ substantially from earlier work
> >
> > It **does not** differ substantially. Muennighoff et at. (cited in the paper) already selects data samples based on difficulty (measured by win rate of an LM) and response length.
> >
> > Moreover, the approach for measuring difficulty used in this work -- asking LM to provide it directly based on the problem formulation -- is biased, and depends on the capability of LM to estimate difficulty without actually trying to solve the problem. This is in contrast to the method used in prior works -- where the LM tries to solve given problem multiple times, and the metric is based on the win-rate.
> >
> > >the improvements are clearly supported by the results
> >
> > The improvements are marginal compared to simple baseline of selecting samples based on length, which doesn't require to run costly evaluation by LM.
> >
> > >under the constrained setting of using only 10% of the data, even small absolute gains translate into meaningful efficiency improvements
> >
> > This hypothetical scenario would occur if we had limited compute budget. And in such scenario it would be more efficient to use simple length-based heuristic, instead of spending additional compute on evaluating difficulty by running inference with LM.
> >
> > >Figure 6 shows that the benefit of Select2Reason remains robust as subset size varies
> >
> > The subsets are still under 19k samples, which is a very low-data regime, not used in practice. Please see the scaling curve of number of prompts vs AIME performance in [1].
> >
> > >Thus, the claim that our approach heavily relies on a powerful teacher is a misunderstanding.
> >
> > I did not claim anything like that. My point was that the metric is biased, and depends on the judge model used. Intuitively, using smaller model will lead to higher bias and less accurate difficulty metric. In prior work, the LM is used to solve each problem multiple times, and based on the total accuracy, we can determine the actual difficulty of problems.
> >
> >
> > ---
> >
> > I will keep my score unchanged.
> >
> > ---
> >
> >  [1] Lu, Kevin and Thinking Machines Lab, "On-Policy Distillation", Thinking Machines Lab: Connectionism, Oct 2025.

---

### Official Review · Reviewer_RqGQ · 2025-11-03

**Soundness:** 2
**Presentation:** 3
**Contribution:** 1
**Rating:** 2
**Confidence:** 3

**Summary:**

This paper proposes SELECT2REASON, a data selection framework for long chain-of-thought (CoT) instruction tuning. The method combines reasoning trace length and question difficulty through a weighted ranking scheme to efficiently choose high-quality examples for fine-tuning. Experiments on multiple math reasoning benchmarks show that training on only 10% of the selected data achieves comparable or superior results to full-data fine-tuning.

**Strengths:**

This work tackles data efficiency for long-CoT reasoning, which is a key bottleneck in current LLM research. It combines reasoning trace length and difficulty in a joint ranker is intuitive.

**Weaknesses:**

1. The paper motivates the heuristics empirically, but does not provide deeper theoretical analysis on why the combination works, lacking theoretical grounding

2. Both “trace length” and “difficulty” metrics are individually known heuristics; their combination, while practical, may be viewed as incremental.

3. Using LLM-as-a-Judge to estimate difficulty may introduce selection bias; this aspect is not systematically analyzed

**Questions:**

1. Could the method be extended to non-mathematical reasoning tasks without redesigning metrics?

2. Does the efficiency advantage hold when scaling beyond 7B models or multi-turn reasoning datasets?

---

> ### Author Response · Authors · 2025-11-19
> **Response to Reviewer RqGQ**
>
> We thank the reviewer for the feedback. Below we provide detailed responses to each point.
>
> ---
>
> > **W1: Lack of deeper theoretical analysis**
>
> We acknowledge the reviewer’s observation. At present, the mainstream and most influential works in this area—including LIMO [1] and S1 [2]—also rely primarily on extensive empirical evidence rather than theoretical guarantees. This is largely due to the nature of the problem: long-CoT instruction selection remains a highly practical and engineering-oriented challenge. The primary purpose of our work is to identify a small yet highly effective subset from a massive instruction pool, thereby reducing the training cost without sacrificing reasoning performance. As demonstrated in *Table 4: Computational overhead of data selection*, the savings are substantial and practically meaningful.
>
> Nonetheless, we agree that a theoretical formulation is valuable. In future work, we plan to explore whether instruction selection can be cast as a formal optimization problem—e.g., maximizing the total influence of chosen samples, where each instruction contributes an individual utility and combinations of instructions may introduce additional synergistic gains. This could provide a theoretical lens under which our heuristic Select2Reason strategy may be analyzed or even justified.
>
> ---
>
> > **W2: Combination of heuristics may seem incremental**
>
>
> While both trace length and difficulty have been explored independently in prior works, the perspective and task setting in our paper differ significantly. Long-CoT reasoning, as emphasized in DeepSeek-R1 [3] and other recent studies, introduces unique challenges that do not appear in short-CoT or standard instruction-tuning scenarios. In these long-chain settings:
>
> (1) Trace length no longer merely reflects verbosity, but correlates with the model’s ability to perform multi-step reasoning, multi-hop decomposition, and reflective correction.
> (2) Difficulty interacts with long-CoT dynamics in ways not observed in ordinary QA or instruction tasks—long reasoning chains tend to amplify both the benefits and the weaknesses of problem difficulty.
>
> Thus, the combination of these two signals is not a trivial stacking of known heuristics, but is motivated by the structural characteristics of long-CoT data. Our extensive ablation studies consistently show that their combination yields meaningful and robust improvements.
>
> ---
>
> > **W3: Potential selection bias from using Qwen2.5-Math-7B as the judge model**
>
>
> Our use of Qwen2.5-Math-7B is motivated by two key factors:
> (1) We follow the setting of LIMO [1], which applies a difficulty filter using Qwen2.5-Math-7B-Instruct for short-CoT reasoning. We improve this by developing an automated difficulty-aware judging model, avoiding the need for expensive human verification pipelines.
> (2) Our base model for long-CoT supervised fine-tuning is also Qwen2.5-Math-7B, ensuring architectural consistency and avoiding unnecessary compute on easy problems that the model can already solve with short reasoning.
>
> As shown in *Table 2*, the same trained difficulty-judging model generalizes well to the *Chinese-DeepSeek-R1-Distill* dataset, a domain and language distinct from the original English math pool. Qualitative analyses in Appendix A.5 further support that this difficulty-aware judge model reliably identifies hard instructions even without domain-specific tuning. These results suggest that the risk of selection bias is limited in practice.
>
> ---
>
> > **Q1: Can the method extend beyond mathematical reasoning**
>
>
> We would like to draw the reviewer’s attention in section *Experiment Results and Analysis*, Line 444, we have already evaluated Select2Reason on **logical inference**, **scientific QA**, and **commonsense reasoning**—all non-mathematical long-CoT tasks. The performance gains remain stable across these domains, demonstrating that the method generalizes beyond mathematics without redesigning metrics.
>
> ---
>
> > **Q2: Does the efficiency advantage hold for larger models or multi-turn reasoning datasets**
>
>
> Due to limited computational resources and development time, our current experiments focus on medium-scale models. Our cross-model evaluations already include Qwen2.5-3B and LLaMA3.1-8B (Line 414), and the observed trends remain consistent. We consider to larger models and broad settings as a next step and plan to explore this in future work.
>
> ---
>
> Thank you for your patience in reading our response. We hope that with a more careful reading you can clarify these misunderstandings and avoid unnecessary debate.
>
> ---
>
> [1] Ye Y, Huang Z, Xiao Y, et al. LIMO: Less is More for Reasoning. COLM 2025.
> [2] Muennighoff N, Yang Z, Shi W, et al. S1: Simple Test-Time Scaling. EMNLP 2025.
> [3] Guo D, Yang D, Zhang H, et al. DeepSeek-R1: Incentivizing reasoning capability via RL. arXiv 2025.

---

### Official Review · Reviewer_tmCD · 2025-11-03

**Soundness:** 4
**Presentation:** 3
**Contribution:** 2
**Rating:** 6
**Confidence:** 4

**Summary:**

This paper proposes the SELECT2REASON framework, which efficiently selects high-quality long-chain-of-thought (long-CoT) reasoning data from large-scale instruction pools by quantifying question difficulty and reasoning trace length. Experiments show that fine-tuning models with only 10% of the carefully selected data achieves performance comparable to or surpassing that of models trained on the full dataset and open-source baselines across multiple mathematical reasoning benchmarks.

**Strengths:**

Methodological Rigor: The framework is simple yet effective, leveraging quantifiable metrics (difficulty scores, trace length) without complex computations.
Comprehensive Experiments: Evaluations span 9 mathematical benchmarks, multiple data scales (2%–10%), and diverse models (Qwen, LLaMA), ensuring reliability.
Reproducibility: Experiments are based on public datasets.

**Weaknesses:**

Metric Interplay Under-explored: The combination of difficulty and trace length relies on a simple weighted sum. A deeper analysis of their correlation or conflict scenarios (e.g., long-easy vs. short-hard traces) would strengthen the method's rationale.
Potential Bias in Difficulty Scoring: The "LLM-as-a-Judge" approach may inherit biases from the specific judge model (Qwen2.5-Math-7B). Using a judge model to filter data might systematically favor a certain type of "difficult problem," while overlooking other equally valuable types of "difficult problems" that are not recognized by the model.
Generalization Scope is Narrow: Strong empirical validation is primarily within mathematical reasoning. Claims about general "long-chain-of-thought reasoning" would be more solid if supported by tests on diverse domains.

**Questions:**

1.The optimal weight (w=0.25) greatly favors trace length over difficulty. What is the underlying reason? Is this ratio consistent across different model scales or data pools?
2.The ablation study shows that diversity-aware selection (Diverse) does not yield significant gains. Given that diversity is often crucial in other data selection paradigms, what is your hypothesis for why it is less critical for long-CoT reasoning instruction tuning specifically? Does this imply that the quality of individual reasoning traces outweighs the benefit of covering a wider range of problem types in this context?
3.When combining rankings, has consideration been given to more complex fusion methods (such as multi-objective optimization) instead of simply using linear weighting? Why do you believe that linear weighting is sufficient to capture the interaction between difficulty and length?

---

> ### Author Response · Authors · 2025-11-19
> **Response to Reviewer tmCD**
>
> We appreciate the reviewer’s thoughtful comments and the opportunity to clarify our contributions. Below we provide detailed responses to each of the points you kindly raised:
>
> ---
>
> > **W1: Metric Interplay Under-explored**
>
> Please refer to **Reviewer SA5m** in > **Q2: Disentangle the effects of trace length and difficulty.** due to page limit.
>
> ---
>
> > **W2: Potential Bias in Difficulty Scoring**
>
> Our use of Qwen2.5-Math-7B is motivated by two key factors:
> (1) We followed the setup of LIMO [1], which “applied a baseline difficulty filter using a short-CoT mathematical model, Qwen2.5-Math-7B-Instruct,” and developed an automated difficulty-aware judging model to replace the cumbersome manual pipeline.
> (2) The base model we use to stimulate reasoning ability is also Qwen2.5-Math-7B, allowing us to avoid wasting compute on problems solvable with short reasoning chains and instead focus on those that truly require long-CoT reasoning.
>
> Moreover, as shown in Table 2, experiments conducted on the *Chinese-DeepSeek-R1-Distill* dataset—using the same trained difficulty-judging model—achieved solid generalization performance. The qualitative cases in Appendix A.5 further illustrate that this model performs well on Chinese data.
>
> ---
>
> > **W3: Generalization Scope is Narrow**
>
> We would like to draw the reviewer’s attention to Section 6 (*Experiment Results and Analysis*, Line 444), where SELECT2REASON demonstrates robust generalization across both domains and model scales. We have now added additional experiments covering *logical inference*, *scientific QA*, and *commonsense reasoning* beyond mathematical exams, and the observed performance gains from long-CoT activation and Select2Reason’s selection strategy remain consistent.
>
> ---
>
> > **Q1: Ratio of Trace Length and Difficulty**
>
> As discussed in Figure 3 of Section 3 (*Preliminary Exploration*), the influence of trace length is indeed significant, which aligns with our choice of a higher weight on this feature. We elaborated that in long-chain reasoning instructions, *rethinking tokens* (e.g., “However,” “Wait,” “Alternatively”) are meaningful indicators of reasoning quality. Similar observations have been made in related studies [3][4].
>
> To further verify this, we used Qwen2.5-Math-72B as a teacher model to generate solutions for the same subset sampled by Select2Reason and fine-tuned Qwen2.5-Math-7B on it. The results are as follows:
>
> | Model          | AIME 24 | AIME 25 | AMC 23 |
> |----------------|---------|---------|--------|
> | Qwen2.5-7B     | 0.090   | 0.106   | 0.558  |
> | Full-Pool      | 0.465   | 0.352   | 0.816  |
> | Select2Reason  | 0.433   | 0.335   | 0.808  |
> | S2R-Qwen-72B   | 0.148   | 0.140   | 0.612  |
>
> The Qwen2.5-Math-72B model, being a short-thinking model, produces reasoning traces that are much shorter and contain fewer reflection tokens than those in OpenR1-Math (statistics omitted due to space). This aligns with our earlier exploration.
>
> ---
>
> > **Q2: Why Diversity Is Less Critical**
>
> Please refer to Reviewer UD9B in > **W1 & Q1:Could the authors supplement the ablation analysis on diversity for long-CoT instruction selection** due to page limit.
>
> ---
>
> > **Q3: More Complex Fusion Methods**
>
> Because difficulty scoring and trace-length computation are handled independently based respectively on *inputs* and *outputs*, and our goal for Select2Reason was to design a *simple yet effective* strategy. The linear weighting approach ensures generalizability and simplicity. The best weight was determined empirically, and from a performance perspective, this empirical linear combination has proven sufficient to capture the optimal balance between the two features.
>
> ---
>
> Thank you for your time and consideration. We will incorporate all additional analyses and clarifications into the revised version as a sign of our appreciation for your valuable feedback. We look forward to your positive response.
>
> [1] Ye Y, Huang Z, Xiao Y, et al. *LIMO: Less is More for Reasoning.* COLM, 2025.
> [2] Guo D, Yang D, Zhang H, et al. *Deepseek-r1: Incentivizing reasoning capability in LLMs via reinforcement learning.* arXiv:2501.12948, 2025.
> [3] Yeo E, Tong Y, Niu X, et al. *Demystifying Long Chain-of-Thought Reasoning in LLMs.* ICLR 2025 Workshop.
> [4] Liu Z, Chen C, Li W, et al. *Understanding R1-zero-like training: A critical perspective.* COLM 2025.

---

### Official Review · Reviewer_SA5m · 2025-11-03

**Soundness:** 1
**Presentation:** 2
**Contribution:** 1
**Rating:** 2
**Confidence:** 5

**Summary:**

This paper introduces SELECT2REASON, a data selection framework designed to reduce the high computational cost of supervised fine-tuning for long CoT reasoning. The authors hypothesize that instruction quality can be effectively proxied by two simple heuristics: the length of the reasoning trace and the difficulty of the question. The proposed method involves training a small "judge" model to assign a difficulty score, and then creating a joint rank that combines this score with the trace length. The core claim is that fine-tuning a model on just the top 10% of data selected by this method can achieve comparable or even superior performance to training on the entire dataset, as demonstrated on the OpenR1-Math-220k dataset with a Qwen2.5-Math-7B model.

**Strengths:**

1. The method relies on two intuitive and low-cost heuristics (trace length and difficulty). The finding that trace length, in particular, serves as a proxy for data quality is a useful, simple baseline.
2. The central finding—that a 10% subset can outperform the 100% full dataset—is compelling and demonstrates a path toward more efficient tuning for complex reasoning tasks.

**Weaknesses:**

1. Validation is limited. The primary claim that 10% selected data surpasses 100% full-pool data is demonstrated on only one model and dataset (Qwen2.5-Math-7B on OpenR1-Math). While generalization is explored, this core efficiency-performance tradeoff is not shown to hold for other model families (e.g., Llama, Mistral). The paper would be stronger if this main result (Table 1) were replicated on at least one other distinct model architecture.

2. Key baselines are not included. The paper discusses alternative data selection methods from prior work, such as using model-specific loss or perplexity (lines 294-295), but fails to include them as baselines in the main experiments. Without this comparison, it's impossible to know if SELECT2REASON is genuinely superior to other established methods for identifying "difficult" or "informative" samples.

3. Questionable difficulty measures. The paper's claim that "Existing work has not established quantitative criteria for identifying the difficulty of questions" (line 211-212) appears to overlook a large body of work that uses metrics like training loss, perplexity, or pass@k as proxies for difficulty. The "LLM-as-a-Judge" method proposed here is not well-justified over these simpler alternatives, and its own reliability and calibration are not thoroughly validated.

4. Confounding variables. The method treats "reasoning trace length" and "question difficulty" as two distinct signals to be combined. However, these two variables are almost certainly highly correlated; a harder question is very likely to require a longer reasoning trace. The paper fails to disentangle these factors, making it unclear if "difficulty" adds any real signal beyond what "length" already provides. The strong performance of the "Longest" baseline (w=0) supports this; the joint ranker's optimal w=0.25 only provides a modest boost, suggesting length is doing most of the work.

5. Unfair comparisons. The comparisons in Table 1 to external models like R1-DISTILL-QWEN and OPENR1-QWEN are not well-controlled, as these models use different data sizes and (as the user notes) likely different pre-training and post-training mixtures. This distracts from the paper's main, valid comparison, which is between the SELECT2REASON subset and the FULL-POOL on the same base model.

**Questions:**

1. The paper presents a counter-intuitive finding: training on long reasoning traces (from the selected data) results in a model that uses fewer thinking tokens (i.e., is more concise) during inference. What is the proposed mechanism for this? Does the model learn to be concise from verbose examples, or does it learn a more robust reasoning process that allows it to find the correct answer more directly, thus requiring less backtracking at inference time?

2. Following on the weakness of confounding variables: Can the authors provide an analysis to disentangle the effects of trace length and difficulty? For example, what is the performance of subsets selected from "long-and-easy" vs. "long-and-hard" vs. "short-and-hard" instructions? This would clarify whether length is merely a proxy for difficulty or if it provides an independent quality signal.

3. Why was the "LLM-as-a-Judge" approach for difficulty chosen over more standard, lower-cost metrics like perplexity or loss from the base model or the pass@k of the base model? Was this novel judge found to be a better predictor of difficulty than these established metrics?

4. Given that many data selection works use training loss or perplexity, how would SELECT2REASON compare directly against a baseline that simply selects the top 10% of data with the highest training loss (or perplexity) when fine-tuned on the full dataset?

---

> ### Author Response · Authors · 2025-11-19
> **Response to Reviewer SA5m**
>
> We appreciate the reviewer’s comments. Overall, **your review contains many misunderstandings**. We kindly ask you to reread the paper in order to clarify the inaccurate points. Below we provide responses to each of the issues you raised:
>
> ---
>
> > **W1: Validation is limited.**
>
> **We are particularly puzzled by the reviewer’s misunderstanding.** In Section “Experiment Results and Analysis,” Line 414 and Line 444, we validate Select2Reason on an additional Chinese-DeepSeek-R1-Distill-data dataset as well as on Qwen2.5-3B and LLaMA3.1-8B. We clearly provide experimental evidence showing that Select2Reason demonstrates strong generalization capabilities by enabling low-cost transfer to other Long-CoT reasoning instruction pools, and that it exhibits robust generalization across domains and model scales.
>
> ---
>
> > **W2 & W3 & Q3 & Q4**: Key baselines not included. questionable difficulty measures. Loss and perplexity-based strategy.
>
> Our work mainly follows and extends the settings used in accepted papers such as LIMO [1] and S1 [2], focusing on common data-quality-oriented filtering features such as question difficulty and diversity to design a better data pipeline. Regarding the strategy of using model-specific loss or perplexity (lines 294–295), this requires token-level computation. We provide a simple cost comparison:
>
> Length statistics (input):
> $\bar{L}_{\text{in}} = 260.27$
>
> $L_{\text{in}}^{\min} = 12$
>
> $L_{\text{in}}^{\max} = 7468$
>
> Length statistics (output):
> $\bar{L}_{\text{out}} = 13085.18$
>
> $L_{\text{out}}^{\min} = 71$
>
> $L_{\text{out}}^{\max} = 77466$
>
> Avg. cost for scoring:
> $C_{\text{score}}^{\text{avg}} = \bar{L}_{\text{in}} + 1$
>
> Avg. cost for perplexity:
> $C_{\text{ppl}}^{\text{avg}} = \bar{L}_{\text{in}} + $
>
> $\bar{L}_{\text{out}}$
>
> Complexity (scoring):
> $\Theta(N \cdot \bar{L}_{\text{in}})$
>
> Complexity (perplexity):
> $\Theta(N \cdot (\bar{L}_{\text{in}} + $
>
> $\bar{L}_{\text{out}}))$
>
> Since output length is much longer than input length, perplexity-based methods are not feasible for filtering long chain-of-thought traces. The same applies to training-loss-based methods, which leads to prohibitive computing costs.
>
> ---
>
> > **W4: Confounding variables.**
>
> The claim that "a harder question is very likely to require a longer reasoning trace" only holds when the model already possesses strong reasoning ability. For publicly available instruction pools collected from the Internet, **we cannot know from an oracle perspective whether a sample with a long reasoning trace corresponds to a difficult question**. When multiple samples exhibit similar reasoning trace lengths, question difficulty becomes an important selection feature. This is exactly the motivation for including difficulty, even though its weight is slightly lower than trace length.
>
> ---
>
> > **W5: Unfair comparisons.**
>
> **Again, we are puzzled by the reviewer’s misunderstanding**. Select2Reason uses fewer instructions for fine-tuning, yet matches or even surpasses R1-Distill-Qwen and OpenR1-Qwen, which use much larger datasets. This is precisely the contribution of our work: with carefully selected high-quality instructions, we can achieve stronger performance than using large-scale data.
>
> ---
>
> > **Q1: A counter-intuitive finding.**
>
> We do not consider this result counter-intuitive. Models trained on high-quality selected data indeed generate fewer reasoning steps on average, as shown in Figure 8 and Figure 12. The case study in Figure 10 also illustrates this clearly: “The emergence of more rethinking tokens in long-CoT reasoning does not lead to improved efficiency. In contrast, stronger LRMs reach the correct reasoning trace more efficiently."
>
> ---
>
> > **Q2: Disentangle the effects of trace length and difficulty.**
>
> | Model          | AIME 24 | AIME 25 | AMC 23 |
> |----------------|---------|---------|--------|
> | Qwen2.5-7B     | 0.090   | 0.106   | 0.558  |
> | Full-Pool      | 0.465   | 0.352   | 0.816  |
> | Select2Reason  | 0.433   | 0.335   | 0.808  |
> | Long & Easy    | 0.398   | 0.290   | 0.771  |
> | Short & Hard   | 0.215   | 0.140   | 0.582  |
>
> We conducted additional experiments to examine the interaction between trace length and difficulty. By independently reversing these two data characteristics and sampling 10% of each as subsets, we obtained the Long & Easy and Short & Hard results. Consistent with our observations in the Preliminary Exploration section, deliberately selecting short traces or easy samples leads to performance degradation, and the negative impact of the former is significantly stronger than that of the latter.
>
> ---
>
> Thank you for your patience in reading our response. We hope that with a more careful reading you can clarify these misunderstandings and avoid unnecessary debate.
>
> [1] Ye Y, Huang Z, Xiao Y, et al. LIMO: Less is More for Reasoning. COLM, 2025.
> [2] Muennighoff N, Yang Z, Shi W, et al. S1: Simple Test-Time Scaling. EMNLP 2025.

---

### Official Review · Reviewer_UD9B · 2025-11-12

**Soundness:** 3
**Presentation:** 2
**Contribution:** 3
**Rating:** 6
**Confidence:** 3

**Summary:**

## Summary
This paper presents SELECT2REASON, an efficient instruction-tuning data selection framework designed to address the high training overhead associated with activating long chain-of-thought (long-CoT) reasoning capabilities in pre-trained large language models (LLMs) via supervised fine-tuning (SFT). While SFT on instruction datasets synthesized by powerful large reasoning models (LRMs)—such as DeepSeek-R1—serves as a cost-effective alternative to reinforcement learning, large-scale instruction sets (e.g., those with over 100k samples) impose substantial computational costs, and automated strategies for long-CoT instruction selection remain underexplored. To fill this gap, SELECT2REASON focuses on two key quantifiable heuristics for identifying high-quality instructions: question difficulty (estimated using a fine-tuned LLM-as-a-Judge) and reasoning trace length (linked to higher frequencies of rethinking behaviors like self-correction and backtracking, which are taken as indicators of strong reasoning quality). These two metrics are integrated through a weighted joint ranking scheme to prioritize high-utility examples.

Empirical evaluations on the OpenR1-Math-220k dataset (with 196k instructions retaining correct answers) show that fine-tuning the Qwen2.5-Math-7B-Instruct model on just 10% of the data selected by SELECT2REASON achieves performance comparable to or superior to full-pool SFT and open-source baselines (e.g., OpenR1-Qwen-7B with 94k samples and DeepSeek-R1-Distill-Qwen-7B with 800k samples) across nine mathematical benchmarks. These benchmarks include three competition-level ones (AIME 2024/2025, AMC 2023) and six comprehensive ones (MATH-500, OlympiadBench, Chinese GAOKAO 2023/2024, GAOKAO Math, KAOYAN). Additional experiments validate the framework’s scalability across different data sizes, its efficiency in reducing inference tokens while preserving performance, and its generalizability to other long-CoT instruction pools (e.g., the 110k-sample Chinese-DeepSeek-R1-Distill dataset) with minimal adaptation costs. The paper also confirms that diversity—a common metric in general instruction selection—does not meaningfully enhance long-CoT reasoning, further highlighting the value of its proposed heuristics.


## Strengths
1. **Clear Problem Definition & Novelty**: The paper addresses a critical, underexplored gap in long-CoT reasoning—namely, efficient instruction selection for SFT to reduce training overhead. Its core insight (prioritizing reasoning trace length and question difficulty as quantifiable heuristics) is well-motivated and novel, filling the void of automated selection strategies for long-CoT data.
2. **Rigorous Preliminary Exploration**: Pre-experiments linking trace length/difficulty to rethinking token frequency (a proxy for reasoning quality) provide a solid empirical foundation for the proposed method. This “motivation → validation” workflow aligns with the standards of top machine learning conferences.
3. **Comprehensive Experimental Design**: The study validates SELECT2REASON across 9 mathematical benchmarks (3 competition-level, 6 comprehensive), multiple model scales (Qwen2.5-3B/7B, LLaMA-3.1-8B), and cross-dataset generalization (Chinese-DeepSeek-R1-Distill). Ablation studies (on hyperparameter *w* and subset size) and cost-benefit analysis further enhance the work’s rigor.
4. **Practical Impact**: Achieving competitive or superior performance with only 10% of the data (compared to full-pool SFT and baselines using 800k samples) delivers tangible value—reducing computational costs and energy consumption, which aligns with the broader trend of sustainable AI development.


## Weaknesses
1. **Insufficient Analysis of Diversity’s Irrelevance**: The paper concludes that diversity does not contribute to long-CoT instruction selection, but its reasoning is underdeveloped. Is this result dataset-specific (e.g., because OpenR1-Math is already sufficiently diverse in mathematical concepts) or an inherent property of long-CoT reasoning tasks? Could combining diversity with trace length/difficulty yield additional performance gains for less curated datasets? No ablation studies (e.g., comparing the joint ranker with/without diversity signals) or qualitative analyses are provided to explain *why* diversity fails to add value in long-CoT selection.
2. **Rethinking Tokens as a Proxy for Reasoning Quality**: The paper links trace length/difficulty to the frequency of rethinking tokens (e.g., “Wait,” “Alternatively”) as a sign of high-quality reasoning. However, it fails to address edge cases: Are there instances where long traces or difficult questions contain few rethinking tokens but still drive significant SFT gains in long-CoT reasoning? Conversely, do low-quality traces (e.g., those with redundant “maybe” statements) artificially inflate rethinking token counts without providing useful supervision? A qualitative comparison of such cases would strengthen the validity of this proxy.
3. **Generalization to Non-Mathematical Long-CoT Tasks**: While the paper includes a small set of experiments on broader tasks (e.g., ZebraLogic, GPQA), its core evaluations focus exclusively on mathematics. Long-CoT reasoning is critical for non-mathematical domains, such as coding (where reasoning involves syntax validation and algorithm design), complex logical inference, and scientific problem-solving. Does SELECT2REASON generalize to these domains? For example, would trace length and difficulty be equally effective for selecting coding instructions (where reasoning quality depends on syntax correctness and semantic alignment rather than mathematical steps)? No systematic cross-domain validation is provided beyond mathematics and limited logical QA tasks.


## Questions
Q1: Could the authors supplement the ablation analysis on diversity for long-CoT instruction selection? (Related to Weakness 1)
Q2: Do edge cases exist in the dataset (e.g., long traces/difficult questions with few rethinking tokens that still boost SFT gains, or low-quality traces with inflated rethinking token counts)? If so, could the authors provide a brief explanation for these cases? (Related to Weakness 2)
Q3: Have the authors conducted any experiments on non-mathematical long-CoT reasoning tasks (e.g., coding, scientific problem-solving)? If yes, could they share the relevant results; if not, could they discuss the potential generalizability of SELECT2REASON to such tasks? (Related to Weakness 3)

**Strengths:**

See Summary

**Weaknesses:**

See Summary

**Questions:**

See Summary

---

> ### Author Response · Authors · 2025-11-19
> **Response to Reviewer UD9B**
>
> We appreciate the reviewer’s thoughtful comments and the opportunity to clarify our contributions.  Below we respond to each of the points you kindly raised:
>
> ---
>
> > **W1 & Q1:** Could the authors supplement the ablation analysis on diversity for long-CoT instruction selection?
> ---
> We appreciate the reviewer’s rigorous perspective on whether **combining diversity with trace length/difficulty** could offer additional gains.  We therefore conducted the following 10% subset ablation experiment (Pass@1 shown below):
>
> | Model| AIME 24 | AIME 25 | AMC 23 |
> |----------------|---------|---------|--------|
> | Qwen2.5-7B | 0.090   | 0.106   | 0.558  |
> | Full-Pool | 0.465   | 0.352   | 0.816  |
> | Longest | 0.425   | 0.312   | 0.805  |
> | Difficult | 0.410  | 0.312   | 0.787  |
> | Select2Reason | 0.433   | 0.335   | 0.808  |
> | S2R-Diverse | 0.364   | 0.292   | 0.766  |
>
> To further understand the effect of Diversity, here is the **problem-type distribution** for the three 10% subsets. Below is the unified table (counts + percentages):
>
> | Method | Geometry | Algebra | Inequalities | Combinatorics | Logic & Puzzles | Number Theory | Calculus | Other |
> |--------|-----------|-----------|----------------|----------------|------------------|------------------|-----------|--------|
> | **Full-Pool** | 40398 (20.56%) | 91714 (46.67%) | 8125 (4.13%) | 18192 (9.26%) | 6605 (3.36%) | 20385 (10.37%) | 9570 (4.87%) | 1513 (0.77%) |
> | **Select2Reason-10%** | 6331 (32.22%) | 3197 (16.27%) | 797 (4.06%) | 4402 (22.40%) | 1097 (5.58%) | 3412 (17.36%) | 314 (1.60%) | 100 (0.51%) |
> | **S2R-Diverse-10%** | 2674 (13.61%) | 2907 (14.79%) | 2488 (12.66%) | 2539 (12.92%) | 2479 (12.62%) | 2553 (12.99%) | 2497 (12.71%) | 1513 (7.70%) |
>
> Our analysis suggests that Diversity-based selection **over-emphasizes tail categories**, unintentionally forcing low-learning-value samples into the subset (cases are not provided due to limited page). Together with the ablations provided above, these results confirm that Diversity **yields negative returns** as a selection signal for long-CoT instruction tuning.
>
> ---
>
> > **W2 & Q2:** Do edge cases exist regarding rethinking tokens, and could the authors explain them?
> ---
> We acknowledge the reviewer’s concern and conduct simple experiments. We randomly selected some reasoning steps and repeated them multiple times to mimic the potential issues of redundancy or cyclical reasoning in the instruction pool for ten and twenty times.
>
> | Model          | AIME 24 | AIME 25 | AMC 23 |
> |----------------|---------|---------|--------|
> | Qwen2.5-7B     | 0.090   | 0.106   | 0.558  |
> | Full-Pool      | 0.465   | 0.352   | 0.816  |
> | Select2Reason  | 0.433   | 0.335   | 0.808  |
> | 10 circular traces  | 0.421   | 0.339   | 0.802  |
> | 20 circular traces    | 0.436   | 0.329   | 0.799  |
>
> Although the experimental results show that Select2Reason appears to be robust to simple circular traces within long-CoT reasoning, these are precisely the reasons why we do **not use rethinking-token frequency as a selection feature directly**. **Human-defined reflection keywords cannot reliably capture the depth of reasoning**. Instead, our method relies only on trace length and difficulty, where two signals are more stable and generalizable. We use rethinking tokens only to validate correlations and causal interpretations, consistent with observations in prior work [1][2][3], rather than as a direct selection metric.
>
> ---
>
> > **W3 & Q3:** Have the authors conducted any experiments on non-mathematical long-CoT reasoning tasks?
> ---
> Actually, we have already evaluated non-mathematical tasks. As described in the section *Experiment Results and Analysis* (line 444), we have already provided cross-domain experiments showing that SELECT2REASON generalizes well beyond mathematics. The evaluation includes logical inference, scientific QA, and commonsense reasoning, and the results are visualized in Figure 9. These experiments demonstrate that our selection criteria remain effective for non-mathematical long-CoT tasks.
>
> Coding tasks require additional development of a code interpreter as a verifier, which introduces extra engineering cost. We acknowledge that this is difficult to implement at the current stage of our work. We will continue to expand our evaluation in the future to assess the performance of Select2Reason on coding tasks.
>
> ---
>
> Thank you for your patience in reading our response. We commit to incorporating all additional analyses and clarifications into the revised version, as a reflection of our appreciation for your valuable feedback. We look forward to your positive reply!
>
> ---
>
> [1] Guo D, Yang D, Zhang H, et al. *DeepSeek-R1: Incentivizing reasoning capability in LLMs via reinforcement learning*. arXiv:2501.12948 (2025).
> [2] Yeo E, Tong Y, Niu X, et al. *Demystifying Long Chain-of-Thought Reasoning in LLMs*. ICLR 2025 Workshop.
> [3] Liu Z, Chen C, Li W, et al. *Understanding R1-zero-like training: A critical perspective*. COLM 2025.

---

### Note · Authors · 2025-12-20

I have read and agree with the venue's withdrawal policy on behalf of myself and my co-authors.